# Navigating digital frontiers in UAE healthcare: A qualitative exploration of healthcare professionals' and patients' experiences with AI and telemedicine

**Azza Alkaabi**[1]*, **Deena Elsori**[2]

1 Department of Student Affairs, Rabdan Academy, Abu Dhabi, United Arab Emirates, 2 Department of Student Affairs, Rabdan Academy, Abu Dhabi, United Arab Emirates

* aalkaabi@ra.ac.ae

## Abstract

The integration of artificial intelligence (AI) and telemedicine into healthcare has significantly advanced patient-centered care, enhancing accessibility, convenience, and patient-doctor relationships. However, different factors determine the extent to which such benefits are realized, especially in unique healthcare settings such as the United Arab Emirates (UAE). In this regard, this research explores healthcare professionals' and patients' perspectives to understand various factors that influence the adoption and use of AI in the UAE's healthcare sector. This research sought to understand the benefits, challenges, and enablers of successful adoption and utilization of AI and telemedicine in the UAE's healthcare setting. Through this objective, this research aims to contribute to the scanty knowledge on the integration of emerging technologies, such as AI, in different infrastructural and cultural contexts. The study employed a qualitative approach, through which eight healthcare professionals and seven patients (totaling 15 participants) were recruited from Dubai- and Abu Dhabi-based hospitals using the purposive sampling strategy. The participants' insights and views on the research topic were captured using semi-structured face-to-face interviews. These interviews were analyzed using the thematic analysis strategy. This study established that while AI and telemedicine are associated with various benefits, including enhancing the management of chronic illnesses, effective controlling of infectious diseases, saving patients and hospitals health-related costs and time, and enhancing convenience, they suffer from various drawbacks, including limited infrastructural and financial resources, significant gaps in skills, safety concerns, and the likelihood of misdiagnosis and misinformation. The study also observed that the successful integration of AI and telemedicine in the UAE healthcare sector necessitated incentivizing stakeholders to use this technology, full involvement and engagement of stakeholders across all stages of implementation, adequate training of the healthcare staff, and public engagement and awareness. This research demonstrates that integrating AI and telemedicine in the UAE healthcare sector necessitates addressing contextual infrastructural and cultural hindrances. The results highlight the need to address such limitations, adequately train healthcare professionals, and enhance data privacy. The study

**Data availability statement:** All relevant data underlying the findings of this study are available within the manuscript and its Supporting Information files, including the anonymized interview transcripts, which have been attached as supplementary materials. These transcripts have been carefully anonymized to ensure participant confidentiality.

**Funding:** The author(s) received no specific funding for this work.

**Competing interests:** The authors have declared that no competing interests exist.

also lays a foundation for further research into contextual challenges hindering the effective adoption and implementation of AI and telemedicine in different healthcare settings in order to develop a generic, context-specific framework that will guide the adoption of such emerging technologies in the global healthcare industry.

## Author summary

Digital health technologies like artificial intelligence and telemedicine are transforming healthcare delivery worldwide. This study explores how healthcare professionals and patients in the UAE perceive and experience these technologies in clinical settings. Through qualitative interviews with 15 participants, we identified both the benefits and challenges of AI and telemedicine adoption. Key findings show that these technologies improve healthcare accessibility, efficiency, and patient engagement. Telemedicine reduces travel burdens and enhances remote monitoring, while AI supports clinical decision-making through data-driven insights. However, challenges such as data privacy concerns, high implementation costs, resistance from healthcare workers, and the need for proper training were highlighted as barriers to widespread adoption. Our research emphasizes that successful integration of AI and telemedicine requires strong policies, infrastructure development, and user-centered training programs to enhance trust and usability. As the UAE continues to invest in digital health, these findings provide valuable insights for policymakers, healthcare institutions, and technology developers aiming to optimize these innovations for better patient care.

## Introduction

Like other sectors, the healthcare sector has witnessed significant technological developments that have revolutionized and improved healthcare provision and the relationship between patients and healthcare professionals [1]. The deployment of advanced healthcare technology in the healthcare sector has significantly transformed the ideology and structure of healthcare management [1]. For instance, the integration of information and communication technologies (ICTs) in the healthcare sector has birthed telemedicine, which enables patients and doctors to interact in real time, hence saving time and costs associated with traveling for medical consultations and examinations [2]. Such technology enhances access to healthcare as it allows patients, even in remote places, to interact with medical professionals from any part of the world.

The effectiveness and efficiency of telemedicine has been enhanced further by the integration of AI; AI has made significant transformations to the healthcare sector by, for example, enhancing diagnosis and treatment accuracy and creating an enabling environment for remote diagnosis and treatment of patients[2]. Therefore, AI and telemedicine have shortened the physical distance that has, over the years, been a great hindrance to access to quality care for patients living in underdeveloped and remote places. The analytic capabilities of AI also mean that enormous amounts of data can be analyzed with relative ease, thus facilitating data-driven or knowledge-based decision-making processes [3]. Although extensive studies have been conducted globally on AI's applications, there remains a lack of research focusing specifically on AI's and telemedicine's impact in the UAE, especially concerning healthcare professionals' and patients' attitudes towards these technologies. While there are

still challenges in its implementation, the technology can immensely and unquestionably transform the healthcare sector, leading to the provision of affordable, quality, and accessible healthcare services [3].

The development of personal computers in the 1990s and the rise of other technological innovations, such as smartphones and social media, significantly influenced the development of the infrastructure required for telemedicine [1,4,5]. These developments resulted in the development of electronic health (e-health), which enhances equal access to crucial health data and information for both patients and medical professionals, leading to better decision-making in healthcare provision. The creation of such technological infrastructure, which allows for telemedicine and AI to thrive, creates an opportunity for the democratization of medical knowledge through the provision of enhanced access to health-related data and the involvement of patients when making decisions involving their health. Therefore, the integration of digital technologies in the health sector restructures various aspects of the healthcare system, including the administration of the treatment and the relationship between doctor and patient, making healthcare provision more patient-centered [1]. Unlike conventional models, technology-based health models also enhance collaboration, partnerships, and communication between medical professionals and patients, leading to personalized treatment and care for each patient and empowerment of patients [1]. It has also provided a mechanism for them to interact with their fellow patients and share experiences [1,5].

Since healthcare providers are critical in the integration of these digital health technologies, their views concerning the opportunities and challenges they present are vital. To them, the transition towards digitalization has created new and complex dynamics in healthcare delivery, considering that patients are gradually appreciating the benefits of such interventions as AI and telemedicine, especially in areas or situations where physical interactions are infeasible. However, the perceptions and attitudes of both healthcare professionals and patients towards these technologies define their true impact. These technologies have not only encouraged proactivity in healthcare delivery but also mitigated the stresses associated with frequent visits to hospitals [5]. Nevertheless, like any other technological intervention, this change has not been free of challenges, most of which are associated with concerns about data privacy violations [5]. This observation suggests the need to understand the contextual factors (barriers and facilitators) affecting digital health technologies' acceptance and implementation. Such an understanding is crucial within the UAE context to maximize facilitators and mitigate barriers effectively.

While a few studies have delved into the adoption and use of digital technologies in healthcare settings, most have employed pretested quantitative models that overlook the unique role of sociocultural and geopolitical settings in influencing the integration of such technologies in unique healthcare settings, such as the UAE. These quantitative studies have overlooked the unique views and perspectives of stakeholders, especially healthcare professionals and patients, which vary depending on stakeholders' experiences and sociocultural and political contexts. The fact that such studies are based on different settings implies that their findings cannot be applied to the UAE context. In this regard, further research is needed to comprehensively understand factors that influence the adoption of digital health technologies, such as AI and telemedicine, in the UAE from the perspectives of the users (patients and healthcare professionals). The present study, therefore, aims to address this knowledge gap by specifically exploring the perspectives of UAE healthcare professionals and patients to understand contextual factors influencing their acceptance, adoption, and use of AI and telemedicine.

By specifically focusing on the UAE healthcare sector, this study will contribute to the global discussion on the integration of technologies in the healthcare sector, giving scholars and practitioners the opportunity to learn more about digital transformation in the UAE

healthcare sector, its success, challenges, and barriers. The new knowledge generated from this research is also expected to help policymakers in UAE and other countries globally identify areas and issues that they should focus on for the successful adoption and implementation of digital technologies such as AI and telemedicine in the healthcare sector. In summary, this study explores (1) healthcare professionals' perceptions and experiences with digital health technologies, focusing on AI and telemedicine; (2) the barriers and facilitators affecting the adoption and effective use of these technologies in clinical settings; and (3) patient experiences and satisfaction levels with AI and telemedicine interventions.

## Methods

### Study design

As mentioned in the introduction, the present study seeks to explore Dubai- and Abu Dhabi-based healthcare professionals' and patients' views and perspectives on the adoption and implementation of AI and telemedicine in the country's healthcare sector based on their experiences. In line with the nature and objective of this study, a qualitative approach was deemed the most appropriate research design. The qualitative research design allows the researcher to interact with the research subjects to capture their views, opinions, beliefs, experiences, and perspectives on the research phenomenon in-depth through semi-structured interviews, focus-group discussions, and observations [6]. It was thus appropriate for this study, which aimed to capture healthcare professionals' and patients' views on the adoption of AI and telemedicine technologies in the healthcare sector. It allowed the researcher to get the subjective perspectives of the participants on this technology, as informed by their knowledge and experiences in the provision of healthcare services, either as providers or clients/recipients. According to Doyle et al. [6], the qualitative research design is suitable for studies based on the "where," "who," and "what" type of questions. Similarly, the present study aimed to understand *what* impacts the adoption of AI and telemedicine has in the healthcare sector, as well as *what* barriers and benefits are associated with such adoption and *who* will be affected. The qualitative approach is widely used in healthcare and nursing research, which is commonly interested in understanding how patients experience a particular intervention; it is interested in understanding the subjects' perceptions and experiences with the research phenomenon, especially in "areas where little is known about the topic"[6]. As mentioned in the introduction, there is scanty knowledge of patients' and healthcare providers' reactions or views on the adoption of AI and telemedicine, particularly in the UAE context. This, therefore, makes qualitative research design the most suitable approach for exploring this research problem.

### Sampling procedure and sample size

Based on the study's objectives and the qualitative research design, a purposive sampling technique was used to select participants. This approach allowed for the selection of individuals with specific, relevant experience in AI and telemedicine, ensuring that participants were knowledgeable and could provide meaningful insights. As Peterson [7] observed, data sources for a descriptive qualitative study are selected purposively, whereby specific considerations must be made before choosing a source. In other words, the researcher recruits only knowledgeable and experienced participants in the subject being studied.

The sample comprised healthcare professionals, patients, and administrators who had direct experience with digital health technologies in the UAE. Participants were recruited from two major Emirates, Dubai and Abu Dhabi, where digital health adoption is more pronounced. This aspect made the two emirates the most relevant settings for exploring this topic in the UAE. Several inclusion criteria were applied to obtain the desired sample population. First, participants had to be

UAE healthcare professionals (doctors, nurses, and telemedicine specialists) or patients who had interacted with AI and telemedicine services. Administrators from healthcare institutions that had adopted these technologies and IT staff involved in their implementation were also considered.

Given the limited resources and the exploratory nature of the study, it was deemed practical to focus on two Emirates rather than extending the study nationally. While this may limit the generalizability, Dubai and Abu Dhabi are representative of the UAE's healthcare advancements in digital technology, hence making them suitable focal points for this research. Referrals and professional networks were used to recruit participants, and only those meeting the specified inclusion criteria were invited to participate. Of the 30 invitations sent, 18 individuals agreed, but three were excluded for not meeting the study's criteria. Ultimately, 15 participants (eight healthcare professionals and seven patients) were interviewed. The demographic characteristics of the participants are summarized in Table 1. This table highlights the diversity in age, gender, education, and experience among the participants, thereby providing context to their perspectives on AI and telemedicine technologies.

**Justification for sample size and data saturation.** The sample size of 15 was determined based on the saturation principle, where additional interviews are unnecessary when new themes or insights cease to emerge [8]. In this study, saturation was observed as the final interviews yielded responses consistent with earlier participants, with no new themes emerging. Although the sample size may appear modest, it aligns with qualitative research norms, where depth of insight is prioritized over breadth [8]. Saturation was confirmed through iterative data analysis, as patterns became repetitive, suggesting sufficient depth in understanding participants' perceptions and experiences.

**Rationale for age range.** Most participants were over the age of 34, with a concentration in their 40s. This demographic profile is representative of experienced healthcare professionals and patients who have had substantial interactions with AI and telemedicine technologies. While younger individuals might have differing views, the age range of participants in this study aligns with the target population in UAE healthcare settings, where senior professionals typically handle the adoption and implementation of new technologies. Future studies could expand to include younger cohorts, but the selected age range was deemed appropriate for the current study's objectives.

Table 1. Demographic characteristics of the participants.

| Participant Code. | Age | Gender | Background |
|---|---|---|---|
| P.1 | 35 | Male | Patient |
| P.2 | 34 | Female | Doctor |
| P.3 | 40 | Female | Patient |
| P.4 | 42 | Female | Patient |
| P.5 | 38 | Male | Patient |
| P.6 | 50 | Male | Nurse |
| P.7 | 56 | Female | Doctor |
| P.8 | 54 | Male | Administrator |
| P.9 | 45 | Male | Patient |
| P.10 | 41 | Male | Administrator |
| P.11 | 54 | Male | Telemedicine Specialist |
| P.12 | 36 | Female | Patient |
| P.13 | 39 | Male | Doctor |
| P.14 | 37 | Female | Nurse |
| P.15 | 48 | Female | Patient |

## Data collection methods

The study relied on semi-structured interviews to collect the necessary data and fulfill the research objectives. Semi-structured interviews were chosen for their flexibility, allowing participants to freely express their experiences and perceptions while enabling the researcher to probe for deeper insights as needed. As mentioned earlier, the study targeted patients, medical professionals, administrators, and IT experts who had interacted with telemedicine and AI technologies in the hospital setting. Since, as mentioned earlier, the study was interested in their experiences and perspectives on these health technologies, data was collected using semi-structured face-to-face and virtual interviews. Before collecting the data using the interviews, the interview questions were subjected to thorough pretesting on a small sample to ensure clarity, relevance, and effectiveness of the interview guide in achieving the research goals. The pretesting stage revealed that some interview questions were ambiguous, while others were unclear. This feedback was taken into consideration and addressed accordingly.

The interviews were administered face-to-face or virtually, based on the participants' preferences; each interview took approximately 60 minutes, regardless the platform used. The interviews followed a defined structure, which not only enhanced logical flow of the interview but also allowed clarification and follow-up questions for deeper probe. The interview guide was divided into two parts. The first part captured the demographic information of the participants, including their area of interaction with AI and telemedicine, age, position/role, and experience (see Table 1 above). The second part captured information pertaining to the research questions and was divided into three sections, according to the three research questions of this study. The first section contained questions capturing practitioners' experiences and perceptions on digital health technologies, in line with the first objective. The second section elicited information on facilitators and barriers of adopting AI and telemedicine, while the last part was interested in patients' satisfaction and experience with telemedicine and AI technologies. The questions included in each section are stated below.

**Section 1: Healthcare professionals' perceptions and experiences with digital health technologies.** In line with the first goal of this study, this section aimed to understand the participants' (healthcare professionals) experiences and perceptions on using AI and telemedicine technologies. The interview questions included in this section are:

(1) "Describe your experiences with using AI and telemedicine in your practice?"

(2) "Why are technologies such as AI and telemedicine used in the healthcare setting?"

(3) "What challenges or barriers do you think hinder healthcare professionals or patients from using or embracing digital health technologies such as AI and telemedicine in healthcare sector?"

(4) "In what ways has the use of AI and telemedicine affected your decision-making, patient interactions, and effectiveness in your practice?"

Though the above questions were used to guide the interview process, additional questions (probe/follow-up questions) were asked, depending on the participant's responses, to capture more details that were needed to comprehensively understand the research problem.

**Section 2: Barriers and facilitators in the adoption of AI and telemedicine.** The second section addressed the study's second objective, which focused on identifying barriers and facilitators affecting the adoption and effective use of AI and telemedicine in clinical settings. The questions in this section included:

(1) "What do you think are the primary barriers to the adoption of AI and telemedicine in clinical settings?"

(2) "What facilitating conditions do you think are needed for the effective use of AI and telemedicine?"

(3) "What advice would you give to an administrator or a healthcare institution seeking to integrate these technologies into their systems?"

These questions were primarily directed at healthcare professionals who directly interact with these technologies and administrators who oversee their adoption and implementation. Probing questions explored deeper aspects of organizational readiness, infrastructural challenges, and the perceived role of training and support in fostering successful adoption.

**Third section: Patient experiences and satisfaction with AI and telemedicine interventions.** The final section explored the study's third objective: understanding patient experiences and satisfaction levels with AI and telemedicine interventions. Questions directed at patients were tailored to explore their personal encounters, insights, and perceptions regarding digital health technologies. The interview questions included:

1) "Describe your experience with AI and telemedicine"

2) How have AI and telemedicine affected your satisfaction with healthcare services?"

3) "What changes or improvements would you suggest to enhance your experience with AI and telemedicine?"

The questions in this section aimed to comprehensively understand whether the patients were satisfied with the adoption of these technologies and suggest areas that should be focused on to enhance their experience and support the use of these technologies in clinical settings. In other words, the questions purposely aimed to evaluate the participants' level of satisfaction with using AI and telemedicine and also identify key areas that management and health department can work on for successful adoption and use of these technologies in the healthcare sector. The structuring of the interview guide based on the research questions eased the process of organizing and analyzing the data thematically.

During the interviews, the participants views were recorded and transcribed verbatim. The interview transcripts were assigned unique identifiers to conceal the identity of the participants, as part of the ethical requirement for this study. Also, during the interviews, the researcher made some notes that were essential during the transcription and analysis of the recorded data as it helped in connecting ideas or themes and also understanding the views better.

## Data analysis

In line with the qualitative approach, the data collected using the interviews was analyzed using the thematic analysis strategy. As the name suggests, thematic analysis entails critically reading the transcribed interviews to identify patterns and themes relating to the research question [9]. Its appropriateness in this study was attributed to its flexibility, which allowed the researcher to identify and categorize interviewees' responses in a manner that effectively addressed the research questions without any theoretical limitations. This flexibility was imperative for this study as it allowed the researcher to identify and organize the participants' views using the three research questions as key thematic areas. In support of this, Braun and Clarke [9] opined that thematic analysis gives researchers the room and space to comprehensively explore the research topic without following any predetermined theoretical constructs.

The analysis commenced with the generation of codes, which were identified from reading and rereading the transcribed interviews, line by line. These codes were then organized into categories, subthemes, and, finally, themes based on the similarity of ideas and concepts

and the frequency of occurrence. The codes, therefore, reflected ideas and concepts that were repeatedly mentioned by the participants and which were relevant to the research questions. A theme was considered significant if it was mentioned in more than 50% of the responses; this was essential in ensuring that the findings reflected the common perspectives of the participants.

Since subjectivity and biases are major problems in qualitative research, the researcher avoided this problem by recruiting two independent researchers who independently coded the data, after which a comparison was made to discuss the differences and reach a consensus. This approach enhanced the credibility of the findings. After a comparison, the final step involved identifying the core themes and subthemes. As mentioned earlier, the core themes were based on the research questions, while subthemes constituted different factors or points that were repeatedly mentioned by different participants during the interviews. The subthemes were related to the pertinent theme/research question and helped in the overall understanding of the participants' perspectives on the use of AI and telemedicine in the healthcare sector.

After coding and developing initial themes, relevant quotes relating to these themes and subthemes were identified to illustrate and support the themes, which enhanced the integrity and credibility of the findings. The inclusion of quotes in qualitative studies helps to eliminate doubts about the possibility of researcher biases in the interpretation and reporting of the data. Furthermore, the coded themes and subthemes were shared with some of the participants to confirm whether they accurately reflected or captured their perspectives, a process known as member checking. This validation process also helped to minimize the risk of misinterpreting the participants' perspectives, hence enhancing the trustworthiness of the findings.

After reviewing and refining the themes and subthemes based on the participants' feedback during the member checking, the next step involved examining the relationship between the participants' responses and their demographic characteristics, such as gender, age, experience, and specialization. The intention of this analysis was to establish whether such demographic characteristics had any influence on the participants' views and opinions on the use of AI and telemedicine. Therefore, the analysis process was robust, as it included several series of coding, member checking, and demographic analysis, all aiming to enhance the rigor, validity, comprehensiveness, and credibility of the study.

## Ethical considerations

The researcher observed several ethical guidelines as this study involved interacting with human subjects. To begin with, the researcher obtained ethical approval, which was issued by the Ethical Committee at the Ministry of Health and Prevention (MOHAP) under the approval number MOHAP/REC/2024/00123. This ethical approval demonstrated the study's compliance with national and international ethical requirements for nursing/healthcare-related research involving human beings. Before participating in the interviews, the participants were issued with an informed consent form, which sought their consent to take part in the study. This form clearly detailed information about the research, including the purpose of the research, data collection procedures, values and benefits of the research and their participation, and their rights. In this regard, they were informed that their participation was voluntary and, hence, they had the right to withdraw at any stage of the interview without explaining or being victimized. They were also informed of their rights to seek clarification for any question or issue. The consent form also assured the participants that their personal details would be anonymized and that their views and opinions would be maintained confidentially in order to protect their privacy. The folders containing their personal data, signed informed

consent forms, and transcribed data were secured in locked cabinets, while the softcopy copies were encrypted with a password to restrict access to unauthorized persons. They were assured that the collected data would be destroyed or deleted after the completion of the research.

Although this is digital health research, the researcher ensured that no personally identifiable information regarding patients was recorded to address ethical concerns associated with privacy. Since, as mentioned earlier, the transcripts would be shared with other researchers during the analysis, it was ensured that all personal details were eliminated during the transcription of the interviews. The interviewees were assigned acronyms to help identify them. After transcriptions, the interviewees were asked to review their transcripts and identify the accuracy of the information as well as any information that they felt would compromise their privacy. Any information that they identified was immediately deleted before data analysis commenced.

## Validation of findings

Member checking was used to validate the findings made from the interviews. This technique involved sharing analyzed data/preliminary results with some of the participants to verify that the findings were accurate, trustworthy, and representative of the views expressed by the interviewees. As such, it improved research credibility by allowing the participants to go through analyzed data and correct possible misrepresentations of their opinions or narratives. While focus groups could have been an alternative approach for member checking, involving multiple participants simultaneously [10], individual follow-ups were chosen to allow each participant the space to provide feedback without potential influence from other participants. The researcher preferred member checking over other validation techniques because the participants were knowledgeable and, hence, helpful in the analysis and interpretation of the data [10].

## Results

Three themes, which aligned with the three research objectives, were identified through the thematic analysis. Therefore, this chapter presents the findings of this study based on the following themes: (1) Perceptions and experiences of healthcare professionals on AI and telemedicine; (2) Facilitators and barriers to the adoption and use of AI and telemedicine hospital settings; and (3) Patient experiences and satisfaction levels with AI and telemedicine interventions.

## Theme 1: Perceptions and experiences of healthcare professionals on AI and telemedicine

The first objective aimed to capture healthcare professionals' experiences and perspectives on digital health technologies, especially AI and telemedicine. Based on the responses, the participants' views were broadly classified into two categories: perceived benefits and challenges.

**Benefits of AI and telemedicine.** *Enhancing patient participation and facilitating patient-centric care:* One of the questions asked when assessing healthcare professionals' perceptions and experiences with AI and telemedicine was about the main benefits and challenges of these technologies in healthcare. This question was motivated by the researcher's curiosity about what healthcare professionals think regarding the benefits of AI and telemedicine. From the data analyzed, participants frequently mentioned that digital health technologies, particularly telemedicine, support a more patient-centric approach, thereby allowing for greater patient involvement in their care. Several participants perceived telemedicine as a platform for sharing responsibilities with patients, which they believed

encouraged patients to engage actively in setting and achieving health goals. One nurse noted, "Digital health technologies such as telemedicine have now made it easier for us to transfer some of the responsibilities to the patients. This way, the patients feel they are participating in efforts to improve their well-being, thus boosting their understanding of why specific decisions have been made." [P.6] This sentiment suggests that telemedicine empowers patients by giving them a role in their care process, which participants felt could improve patient outcomes. Additionally, by assigning some responsibilities to patients, healthcare professionals reported being able to allocate more time to those with critical needs, enhancing overall care efficiency.

In contrast, AI was perceived as having a more indirect role in patient engagement, with participants viewing it primarily as a tool for data analysis and monitoring. Although AI was not as commonly associated with direct patient interaction, some participants thought it could support patient-centric care by generating personalized insights based on patient data. Physicians also highlighted telemedicine's potential to foster proactive, informed patients. Participants felt that telemedicine allowed for more intensive communication, thus enabling professionals to gather vital patient information before in-person appointments. This was seen as enhancing the doctor-patient relationship. One doctor explained, "Incorporating telemedicine in our facility has allowed us to interact with the patients more and understand their needs. Consequently, the doctor-patient relationship improves, thus increasing the possibility of achieving the desired outcomes." [P.2] In summary, participants generally viewed telemedicine as a facilitator of patient engagement, while AI was seen more as a supportive tool that complements patient care through data insights. This distinction highlights the complementary roles of these technologies, with telemedicine fostering direct patient interaction and AI contributing to data-driven decision-making.

*Cost and time saving:* AI and telemedicine were also associated with cost and time-saving. Many participants believed that digital health technologies could reduce the duplication of roles and inefficiencies within healthcare systems, ultimately decreasing associated costs. Additionally, they felt that these technologies allow patients to avoid unnecessary procedures, thus saving productive time. One of the doctors interviewed observed, "Telemedicine has been a revelation in my work because it has enhanced my ability to make accurate diagnoses and identify the most effective ways to manage different cases. Collaborating with patients through telemedicine has also improved my communication and interpersonal skills. Beyond these benefits, telemedicine provides a mechanism for analyzing large quantities of data in real-time at a lower cost compared to traditional methods." [P.13] This statement reflects the view that telemedicine not only aids in improving healthcare professionals' productivity but also minimizes costs by reducing travel-related expenses for patients and streamlining data analysis.

While telemedicine was more commonly associated with direct patient interaction and travel-related savings, participants also viewed AI as a valuable tool for handling data processing, which they believed could enable quicker decision-making and lower operational costs. Several participants highlighted that AI's ability to automate repetitive administrative tasks and streamline data analysis could improve workflow efficiency, which would ultimately reduce the burden on healthcare professionals. Automating routine processes with AI enables clinicians to dedicate more time and attention to complex cases that require their expertise, thereby enhancing the overall quality of care. A healthcare administrator noted, "AI has the potential to handle a lot of the routine, time-consuming tasks that don't necessarily need a human touch. With AI processing patient records and analyzing patterns, our doctors and nurses can focus on what truly matters, providing personalized and attentive care to patients with complicated cases." [P.8] This comment reflects a commonly held view among participants that AI, while not directly involved in patient interactions, plays a

supporting role that can optimize the healthcare system's efficiency and allocate resources more effectively.

*Controlling the spread of infectious diseases:* Another recurring theme in the participants' responses is that using AI and telemedicine can help control the spread of contagious diseases, especially during pandemics such as COVID-19. In particular, healthcare professionals believe that these technologies facilitate remote care (hence allowing patients to receive preliminary assessments from home) and reduce the risk of spreading infectious diseases in healthcare settings. One healthcare provider shared, "The reason I support the integration of AI and telemedicine is their potential to help control the spread of transmissible diseases. These technologies enable patients to get care in the comfort of their homes, reducing the likelihood of exposure to infectious illnesses in crowded hospital settings." [P.13] This view underscores the importance of remote care for infection control, especially for vulnerable populations such as the elderly or those with chronic illnesses.

According to the interviewees, telemedicine helped in remote consultations, ultimately reducing the need for personal visits to clinical settings during outbreaks. Reducing personal visits to healthcare professionals was viewed as one way of controlling the spread of infections, especially in crowded healthcare facilities. The participants argued that integrating telemedicine can help doctors remotely evaluate the patient's symptoms, make prescriptions, and monitor the patients without necessarily meeting face-to-face; this also protects the patients and healthcare professionals, especially during a pandemic. On the other hand, the respondents cited AI as a pivotal tool for monitoring and analyzing the medical data of different patients, thereby helping identify potential outbreaks early enough. Through aggregation and comparison of data from different patients in real-time, the participants noted that AI can efficiently identify patterns in infections and raise the alarm of potential outbreaks early enough. As noted by one of the doctors [P.2], AI has the potential to detect symptom clusters or anomalies in different regions to identify disease hotspots, which can lead to an immediate response by the medical team, hence controlling the disease before spreading to other areas. P.2 indicated, "AI has the potential to track patients' data from multiple regions and establish patterns that signal the possibility of an outbreak, providing crucial insights that can lead to faster and timely isolation of cases before the spread." P.2's views demonstrate the significant role that AI can play in epidemiological surveillance and the development of a responsive healthcare system for infectious diseases.

*Boosting primary care and chronic illnesses management:* A number of participants cited AI and telemedicine as significant boosters of primary care and chronic illness management without necessarily having physical patient-doctor contact. The respondents noted that telemedicine helps in the maintenance of practitioner-patient engagement, which is vital for the management of chronic conditions and the promotion of community health. In particular, P.13 asserted, "Telemedicine eases the interaction between the patient and primary healthcare experts [which] promotes the community health and makes it easier for the monitoring of people with chronic illnesses [thereby] enhancing their quality of life." P.13's sentiments demonstrate the importance of telemedicine in improving community health practices and quality of life, especially among chronically ill patients.

The doctors also associated telemedicine with enhancing the accessibility of health services to patients who need regular monitoring but face mobility challenges or live in remote locations; it helps healthcare professionals tailor the treatment plans according to the unique nature and needs of patients, hence reducing inequalities in access to healthcare. Another participant asserted, "Sometimes mobility is a major challenge for our elderly patients, many of whom need regular monitoring and care. So, telemedicine helps us connect with our patients more often, helping us identify health issues before they escalate [hence] reducing disruptions

to their daily lives." [P.10]. These sentiments demonstrated the importance of telemedicine in providing continuity of care and proactively responding to health issues before escalating, hence improving the quality of life for patients in remote places with serious mobility challenges.

Similarly, the responses from the interviewed participants showed that AI can play a pivotal role in the management of chronic illnesses by monitoring the patient's condition and providing insights using data analytics. The participants added that AI has superior data analytic capabilities, which can be used by healthcare professionals to monitor data trends for different chronically ill patients, identify changes in their health status, predict the outcomes, and alert the providers for timely response. In support of this view, one of the participants, a health administrator, indicated, "AI is capable of analyzing a patient's historical data and alerting doctors when something is off-track. Such features enhance our proactiveness in our care, as treatment plans are adjusted accordingly before deterioration of minor, manageable issues" [P.7]. The above sentiments demonstrated the significance of AI in developing a more responsive healthcare system characterized by data-driven decisions that are customized to meet the patient's needs.

To sum it up, the above insights demonstrate the complementarity of AI and telemedicine in the provision of primary care and management of chronic diseases. The sentiments shared by the participants demonstrate that telemedicine plays a significant role in enhancing patient-doctor interactions and the provision of continuity of care, while AI helps to evaluate health trends and use data analytics to make informed decisions that improve the outcomes of healthcare interventions. Therefore, the integration of the two digital health technologies into the healthcare system enhances clinical efficiency and response, especially to patients with chronic illnesses, ultimately enhancing their quality of life.

**Challenges/drawbacks of using AI and telemedicine.** The interviews also revealed various challenges or drawbacks associated with the use of AI and telemedicine. These drawbacks include misdiagnosis and misinformation, safety issues, and financial challenges.

*Misinformation and misdiagnosis:* Even though AI and telemedicine were associated with the improvement of the healthcare system, some participants cited risks of misinformation and misdiagnosis as major concerns with this digital health technology. The participants indicated that the absence of physical face-to-face interactions with patients in telemedicine increases the risk of misdiagnosis due to inaccurate patient assessments and communication challenges. The lack of physical doctor-patient interactions implies that some aspects of the examination, which are crucial in clinical decisions, such as non-verbal cues like physical language and body language, cannot be conducted, consequently affecting the accuracy and comprehensiveness of such technology-mediated clinical examinations. According to P.11, a telemedicine specialist can significantly benefit the healthcare system when appropriately configured and used, but when misused, "it can negatively impact the relationship between healthcare professionals and patients. It lacks physical face-to-face interactions between patients and doctors, creating room for misinformation and misdiagnosis due to lack of some crucial information that requires physical examination."

Regarding the use of AI, some participants argued that while it is effective in analyzing huge datasets, it lacks human intuition, which is crucial when making ethically complex clinical decisions. This increases the likelihood of AI making diagnostic errors because of over-relying on AI algorithms and being inconsiderate of the patient's unique context. The participants noted that the effectiveness of AI is limited, especially in the diagnosis of rare or atypical conditions, which the algorithm has not been trained on. As explained by one physician, "AI is good at doing routine tasks but can't replace the need for clinical judgment. If the algorithm is trained on flawed or incomplete data, the diagnosis will be terribly flawed,

which is a big clinical risk or error that no physician would want to commit, especially when dealing with patients with complex health issues that require well-thought personalized care and approach." [P.13] These views underpin the importance of balancing AI and telemedicine technologies with traditional diagnostic methodologies to ensure that medical professionals' oversight is central in the provision of care.

*Safety concerns:* Most of the participants cited security as a major concern in the implementation of AI and telemedicine. In particular, many health professionals interviewed in this study indicated that the programming of AI-based health systems to act like human beings increased their susceptibility to human biases and errors, which can lead to deadly mistakes. Therefore, they believed that AI systems should not be entirely relied upon when making clinical decisions because such errors can lead to life-threatening outcomes for patients. One administrator interviewed in this study asserted, "One of the major concerns with the implementation of AI-based systems in the clinical settings is safety. Since these AI algorithms are designed by humans, the designers' (human) biases can be replicated in these algorithms, which can lead to life-threatening errors. From where I stand, we have a long way to go before AI becomes fully implemented in the healthcare sector. To realize its full potential, issues such as safety and data privacy need to be comprehensively addressed to earn the trust of doctors and patients." [P.10] This view highlights the broader concern of users mistrusting AI because of fears that the algorithms might be compromised, leading to lethal mistakes when healthcare professionals solely rely on such systems to make sensitive clinical decisions.

Some participants also cited the stringent regulatory environment as another safety concern associated with using AI and telemedicine in the healthcare sector. It was learned that strict data integrity and safety regulations, while crucial for protecting patients' safety, hindered innovation and creativity and, consequently, the adoption of these digital technologies in healthcare settings. Participant P. 14, a healthcare provider, argued, "AI and telemedicine have one of the strictest regulations, which hinder their adoption and use to their fullest capacity in the healthcare settings. Though these rules aim to safeguard the patients, they, on the other side, limit the extent to which we can fully adopt and exploit the capabilities of this technology to improve patient care." [P.14] This view demonstrates how rigid and strict regulations limit the extent to which healthcare providers can use AI and telemedicine technologies to enhance the quality and efficiency of patient care.

In this regard, the above views demonstrate the complex yet sensitive balance between safety concerns and flexible and effective use and adoption of AI and telemedicine in healthcare settings. The findings show that while safety is pivotal in the adoption and deployment of AI-based and telemedicine systems, strict regulations and the likelihood of using biased AI algorithms in clinical settings hinder the optimal use of these digital health technologies. The findings thus suggest that improvement of regulatory frameworks and addressing the human biases in coding and training AI algorithms can enhance trust and accelerate the adoption and use of digital health technologies such as AI and telemedicine in the healthcare sector.

*Unsuitability for emergency care:* A number of participants raised concerns over the application of AI and telemedicine in emergency situations, arguing that it was unsuitable for such a scenario. They noted that critical emergencies required immediate, hands-on medical interventions, which were lacking in telemedicine; telemedicine was viewed unsuitable for certain lifesaving operations that require medical professionals to be physically present, such as administering intravenous medications, surgery, and physical resuscitation, which require human skills. These views were echoed by P.13, who opined, "Not all medical procedures will work for telemedicine; some needs are life-threatening which require the physical presence of a healthcare professional as they cannot be conducted through a screen. How can some procedures like surgery and resuscitation be administered via a screen?" These sentiments

demonstrate that telemedicine does not work in all scenarios; some situations need a trained healthcare professional to be physically present.

Other participants cited delays as another factor that renders telemedicine unsuitable for emergency situations. For instance, P.11 indicated that telemedicine cannot be used in scenarios where diagnostic tests, such as imaging, laboratory tests, and even physical assessments, are needed immediately. Based on their experience, some participants also noted that some patients might unintentionally fail to provide some crucial information during remote consultations. Combined with the lack of physical contact between the patient and the doctor, chances of misdiagnosis can occur, which can be life-threatening, especially in emergency situations where patients are in pain or unable to effectively express themselves. This was supported by P.11, who argued, "Emergency situations need more in-person evaluations that should be done promptly to save a life. Lack of physical observation, a key characteristic in telemedicine, may result in the omission of crucial details that inform the clinical decisions, leading to the wrong treatment" [P.11]. This statement shows the importance of physical/in-person interaction between the doctor and patient, especially in emergency situations, where time and accuracy of the diagnosis are crucial to the patient.

*Financial constraints:* Many of the participants also mentioned financial costs as another major hindrance to the adoption and use of AI and telemedicine in the UAE health sector. The participants argued that the substantial financial costs associated with the initial implementation and integration of AI and telemedicine technology make it difficult for small, resource-constrained healthcare institutions to implement it. Some noted that implementing such technology necessitates trained personnel, significant expenditure on the hardware and software components of the system, and other logistics that many smaller-sized healthcare institutions, which are crucial in the provision of primary care, cannot afford. According to one of the doctors interviewed in this study, "implementation costs for AI-enabled healthcare system that also integrates AI can overwhelm smaller institutions [which] lack adequate financial resources." [P.7]. Thus, the above sentiments demonstrate high implementation costs for AI and telemedicine prevent many healthcare institutions from adopting it.

Some participants also cited inconsistency in insurance coverage for telemedicine services as another financial barrier to the adoption of telemedicine in many UAE healthcare facilities. The participants noted that some insurers fail to reimburse for telemedicine services offered, which forces patients to use their money to pay for such services. This promotes inequality in access to health care as some patients come from low socio-economic backgrounds where they cannot get such money to pay for their telemedicine services. In support of this view, participant P.10, an administrator, opined, "Insurer's failure to cover telemedicine services forces patients to pay for themselves, some of whom are struggling financially." Such patients might avoid such services, even if they critically need them to manage their conditions, ultimately resulting in a decline in quality of life. The above sentiments, therefore, demonstrate the need to insure telemedicine services to enhance their access, especially to the vulnerable populations.

## Theme 2: Barriers and facilitators affecting the adoption and effective use of AI and telemedicine clinical settings

The second objective of this study was to explore the facilitators and barriers to the adoption of telemedicine and AI. This section presents a number of barriers and facilitators identified from the interview responses.

**Barriers.** *Dealing with voluminous data and strict regulations:* Several participants mentioned the large amounts of data emanating from the use of AI and telemedicine as a

significant barrier to the adoption and effective use of AI and telemedicine in the healthcare sector. The participants argued that employment of this technology creates voluminous data, which most clinical facilities lack adequate infrastructure to store and effectively utilize in making data-based clinical decisions. One of the physicians asserted, "Implementing AI and telemedicine in the UAE will automatically result in the creation of huge amounts of data. I don't think our healthcare system is fully prepared to handle and utilize such volumes of data effectively. This implies that even the benefits associated with the implementation of such technologies will not be realized because AI thrives on data, and if that data cannot be effectively used to make clinical decisions, it will not serve any good to the patient and medical professionals. Major institutional preparation and infrastructural development are needed to ensure that such technological developments benefit both the patients and doctors." [P.7] This perspective highlighted the need for institutional reforms and infrastructural development before the deployment of health technologies to ensure they optimally benefit the intended users and improve the quality of care.

In this study, handling huge amounts of clinical data was linked with data protection and privacy concerns. All participants, including patients, expressed skepticism regarding the ability of existing security mechanisms to protect their personal information from breaches and misuse. Participants felt that if data protection issues were not adequately addressed, it could negatively impact the adoption and trust in AI and telemedicine. One patient shared their concerns, stating, "These technologies will only be widely accepted if they address the fears associated with data protection. Users need to feel that their digitally collected data is secure and not used for other purposes besides treatment or care provision to the patient. The negative assessment of such technologies as AI and telemedicine is based on the fear that sufficient mechanisms have not been put in place to safeguard patients from breach and misuse. If such mechanisms are developed, I believe the skepticism around these technologies would be addressed." [P.1] This quote reflects the heightened sensitivity surrounding data privacy, and it underscores the importance of establishing robust data security frameworks to foster trust and acceptance.

*Infrastructural and technical shortcomings:* Participants noted that, despite the UAE's advanced digital infrastructure, several infrastructural and technical barriers continue to impede the widespread adoption and effective use of AI and telemedicine in clinical settings. One of the primary issues highlighted was the technological incompatibility of AI and telemedicine with the existing workflows of healthcare professionals, particularly doctors. The responses indicated that implementing AI and telemedicine requires healthcare professionals to adjust accordingly, hence causing some disruption. Such disruption was associated with interfering with the effectiveness and productivity of physicians, especially those attending to a high number of patients. Introducing new technologies was associated with adding more complexities to the process, ultimately affecting the performance and productivity of healthcare professionals since significant time would be spent in understanding and integrating the new systems into their clinical practices. One of the administrators asserted, "It's not just about having the technology; it's about making sure it fits into our daily practice. If using AI or telemedicine adds more steps or slows us down, it becomes more of a hindrance than a help." [P.7] Such comment demonstrates that integrating new technologies in the healthcare sector can negatively influence the productivity of medical professionals because of adaptational challenges.

The participants also cited the lack of standardization and limited capacities of smaller health institutions as significant barriers to the adoption of AI and telemedicine. The responses showed that many smaller health facilities, though willing to adopt and implement such technologies, lack the financial strength and technical capabilities to actualize such

visions. This leaves such systems for big, well-funded health institutions, consequently creating disparity in the integration of digital health technologies and quality of care in the country. One of the administrators interviewed in this study explained, "Though UAE has been a global leader in integrating emerging technologies such as AI and telemedicine to enhance its services to people, operationalization of such technologies is facing a major challenge. Some healthcare systems are grappling with the problem of insufficient capacity and lack of standardized systems, which need to be critically relooked to enhance the seamless integration of these technologies. Lack of financial and technical capacity in some institutions only exacerbates inequalities in healthcare access and even makes such technologies a reserve for the well-established, well-financed healthcare institutions." [P.10] These sentiments demonstrate the need for standardization of digital health technologies and enhancement of collaboration among key stakeholders in the healthcare sector to enhance the accessibility of such technologies to healthcare facilities, regardless of their financial and technical capacities.

*Internal resistance to change:* Many participants, especially administrators, noted that internal resistance to changes associated with the implementation of new technologies such as AI and telemedicine is a major concern among healthcare professionals and even some patients. The participants noted that many healthcare professionals are likely to resist any change that has the potential to disrupt their daily routines, workflows, and clinical practices. This resistance stems from fear of over-reliance on such automated systems in decision-making, ultimately affecting their value and roles in healthcare provision. One healthcare professional asserted, "Obviously, AI and telemedicine are disruptors of our usual clinical practices and routines. Many medical professionals may not like such changes and hence oppose them. Some feel that it takes away their role and value as medical practitioners." [P.10] These sentiments demonstrate that fears over losing autonomy controlling clinical practice and decision-making can cause internal resistance to the adoption of AI and telemedicine.

Some participants suggested various strategies that healthcare institutions can use to address such challenges. Some suggested that establishing a digital transformation department in healthcare facilities that exclusively focuses on preparing the institutions and professionals to adopt new technologies can greatly counter such resistance. Others noted that such departments should also play the role of educating medical professionals on such technologies and provide the necessary support to enhance their adaptation to such technological developments. Participant P.11 affirmed, "It is possible to mitigate fear and resistance to change through the creation of a dedicated department that focuses on training, creating awareness, supporting, and adequately preparing healthcare professionals to face such changes." This statement underscores the importance of establishing a structured framework for the adoption of new systems in healthcare settings to minimize resistance and fear among users of such technologies.

*Skills gap:* Lack of requisite skills to effectively work with AI and telemedicine technologies was also cited as a major barrier to the successful adoption and implementation of this technology in the healthcare sector. The participants believed that many healthcare professionals lack technical knowledge and skills in AI and telemedicine technologies, which increases their likelihood of resisting such technology when introduced. These skills gap also affects their ability to optimally use this technology in their practice. This underscores the need for additional training to enhance the healthcare workers' confidence in new technologies. One participant noted, "The widespread adoption and effective use of AI and telemedicine in the UAE healthcare system will automatically increase the demand for tech-savvy employees, especially healthcare professionals, because they are central to their implementation and use." [P.13]. This view demonstrates the importance of adequately preparing medical professionals

through specialized training programs to enhance their technical capacity to effectively work with such crucial digital health technologies.

Besides enhancing healthcare workers' technological skills, participants also highlighted the importance of prioritizing medical professionals with some background knowledge in data analytics or digital health systems. Such professionals are more likely to embrace and adopt new technologies such as AI and telemedicine when introduced in the workplace, hence reducing the resistance and amount of resources spent in changing their attitudes and upskilling their technological capacities. However, the participants noted that training healthcare professionals on digital competencies is inevitable, even when they have such background, because these systems keep evolving, hence the need for healthcare organizations to take proactive approaches to ensure that their employees' technical skills are regularly updated and relevant. This was emphasized by one of the participants, who noted, "More training is needed to bridge the existing gap [because] lack of such training will lead to the underutilization of such technologies and limit their potential in improving the quality of care provided in clinical settings." [P.10] Therefore, the above views demonstrate the need for continuous, structured technological training programs aiming to create a workforce with the relevant technological skills for optimal performance of human-technology hybrid in healthcare settings.

**Facilitators.** Besides understanding the various barriers impeding the adoption and use of AI and telemedicine in healthcare settings, the interview guide included questions that aimed to explore key facilitators of this technology. To this end, the participants identified various AI and telemedicine adoption facilitators, including heightening public awareness, comprehensive training and education for healthcare professionals, active involvement of healthcare providers in the development and implementation processes, and financial incentives to encourage technology adoption. According to the participants, prioritizing these facilitators could enhance the acceptability, adoption, and use of AI and telemedicine, hence creating an integrated healthcare system.

*Awareness of AI and telemedicine:* Public awareness was mentioned as a key factor and enabler of the adoption of AI and telemedicine in the healthcare sector. According to the participants, educating healthcare professionals on the benefits and use of AI and telemedicine can encourage them to embrace such digital health innovation and address fears and misconceptions, leading to increased adoption and use. According to the participants, healthcare professionals are more likely to embrace and adopt technologies that enhance their decision-making capabilities, reduce routine workloads, and improve their clinical practice. One participant shared, "Among the many reasons people resist change is a lack of awareness regarding the benefits of the proposed change. To this end, I believe that increased awareness is one of the key facilitators of increased adoption and effective use of these technologies. All stakeholders, including patients and clinicians, must understand the benefits or usefulness of incorporating AI and telemedicine in clinical settings for them to embrace and accept them." [P.2]

Participants also emphasized the importance of public awareness campaigns to educate patients on the benefits of telemedicine, which they felt would foster trust and reduce skepticism. They suggested that awareness efforts could demonstrate how telemedicine improves convenience and access to healthcare, particularly in rural areas or for patients with limited mobility, while AI could be showcased for its analytical capabilities in supporting diagnosis and treatment. This increased patient awareness could build a foundation of trust, thus reducing potential resistance to digital health interventions. Another participant explained, "Patients also need to understand how these technologies work and how they can benefit from them. They will definitely embrace innovations that enhance efficiency and convenience." [P.4] Therefore, heightening awareness among healthcare professionals and patients can help avert many obstacles to the adoption and use of AI and telemedicine.

***Thorough training and education for healthcare professionals:*** The interviews conducted revealed that the education and training of healthcare professionals catalyzed the adoption and effective use of digital health technologies in the healthcare setting. The participants associated training with the enhancement of healthcare professionals' feelings of self-efficacy, confidence, and perceived usefulness, hence the likelihood of embracing and adopting new technologies in their clinical settings. Many agreed that training boosts confidence, competence, and optimism in new technological innovations, hence changing their negative views about such technologies. One participant opined, "Training is a fundamental component of implementing new technology; it helps users remove doubts and misconceptions they may have about the technology and adequately prepares them to use the new system effectively. It enhances our confidence, feeling of self-efficacy and competence, and believing in our ability to effortlessly undertake various tasks using them." [P.14] These sentiments show that training accelerates the adoption and use of AI and telemedicine in the healthcare sector.

Even though training is instrumental in the adoption and use of new technologies, the participants noted that such training should be flexible enough to accommodate the tight schedules and daily routines of healthcare professionals. As such, participants recommended that healthcare organizations should establish training programs and adjust work schedules to allow healthcare providers ample time to attend these sessions. As explained by one of the participants, the provision of adequate support and time for training ensures that healthcare professionals adequately engage and understand the new technologies to avoid medical errors that may have devastating impacts on the patients. The participant explained, "Unfortunately, many healthcare workers do not have adequate time outside their work schedule to engage in training. This implies that the training program must have a flexible schedule to accommodate their tight schedule and routines while at the same time ensuring that they grasp all knowledge needed to effectively operate such systems to avoid making medical errors that could have life-threatening impacts on patients." [P.13] These sentiments underscore the importance of organizational support in establishing flexible training and educational programs that allocate adequate time for understanding all concepts of the new technology.

***Healthcare providers' engagement in the development and implementation of digital health technologies:*** Collaboration between different stakeholders in the healthcare sector, including healthcare providers, was cited as a critical facilitator of the adoption and use of digital technologies in the healthcare sector. It was established that involving clinicians and other key primary stakeholders from the early phases of the system design and development process helps in understanding the different needs of the end users, hence resulting in the development of a system that resonates with their unique contexts. According to some participants, such a collaborative environment ensures that the new system perfectly aligns with the healthcare professionals' workflows, reducing disruptions and improving efficiency. The participants noted that professionals are more likely to embrace and adopt a system they actively participated in its development as they feel they are part of it. This was echoed by one of the healthcare providers, who argued, "The involvement of healthcare professionals in the development of AI-based systems and telemedicine infrastructure helps to avert some infrastructural and technical barriers that hinder the adoption and use of such systems because it leads to the development of systems that reflect their unique needs and expectations." [P.13]

The participants also believed that the involvement of healthcare providers in the development of digital health technologies such as AI and telemedicine assures healthcare professionals of their autonomy and control over their work, hence boosting their feelings of empowerment and confidence in the new system. Some participants noted that engaging healthcare practitioners enhances a sense of ownership of the process and product, hence viewing themselves as co-creators rather than mere users. One of the interviewees argued,

"Our inclusion in the development process makes us feel like partners and co-creators rather than passive users. Such partnership even leads to a system whose functionality aligns with our unique practices, needs, values, and beliefs. We feel the ownership of the process and product" [P.11] From this statement, it was evident that collaboration in system development helps to avoid many hindrances and create trust and a sense of ownership among the end users.

*Incentivizing the use of digital health technologies:* Several participants associated incentives with the adoption and effective use of AI and telemedicine. The participants believed that providing incentives (whether support programs or financial incentives) encourages healthcare professionals, institutions, and even patients to embrace digital health technologies. They argued that incentives, which could originate from government or non-governmental organizations, seek to address particular barriers or challenges hindering the adoption and effective use of digital health technologies, ultimately encouraging more people or institutions to adopt them. Examples of incentives mentioned by participants include grants or subsidies to reduce implementation costs and free or subsidized training and support programs. In line with this, one of the participants indicated, "Incentives are an indispensable part of the successful implementation and use of new technologies. Users need to feel that the government is supporting them by providing incentives that tackle challenges that would make them reluctant to adopt such technologies." [P.10]. As per this view, incentives can significantly motivate users and institutions to embrace innovations.

According to the participants, incentives do not have to be necessarily monetary. They can be in the form of career advancement opportunities, training, recognition and awards. For instance, the participants argued that non-monetary actions such as recognition of professionals or institutions that have successfully integrated technology into their practice can motivate individuals or other healthcare organizations to implement such innovations, hence leading to the creation of a culture of digital innovation in the healthcare settings. This finding was echoed by participant P.9, who asserted, "Recognition and appreciation for innovations encourages and instills a culture of innovation in healthcare settings. It is a non-monetary incentive that is usually ignored but very important." [P.9] This sentiment therefore demonstrates the significant role tangible and intangible incentives can play in encouraging and accelerating the adoption and effective use of AI and telemedicine in the healthcare sector.

## Theme 3: Patient experiences and satisfaction levels with AI and telemedicine interventions

The third objectives aimed to explore the participants' (patients) experiences and satisfaction with the AI and telemedicine. The objective also sought the participants' views and suggestions on the use of technologies to improve patients' clinical experiences. Based on their responses, several sub-themes were identified, including convenience, enhanced frequency and quality of communication, improved patient-doctor relationships, proactivity in care delivery, and quick discharge of patients.

**Convenience and accessibility of care.** Participants, particularly patients, associated AI and telemedicine with significant improvements in convenience, primarily due to the ability to receive 24-hour care from the comfort of their homes. They felt that these technologies enable remote monitoring, reducing the need to travel to healthcare facilities and alleviating many of the physical and logistical challenges associated with in-person visits. This convenience was seen as beneficial for patients with mobility issues or those in delicate health conditions, as it allows them to receive care within familiar surroundings. One patient explained, "The most significant benefit I can associate with telemedicine is that it makes the process of seeking healthcare seamless and convenient. It enhances the monitoring of

my condition without me having to travel to visit the doctor. This healthcare delivery model is crucial, especially in managing pandemics such as COVID-19, where physical contact was highly prohibited from controlling the spread." [P.1]. This view shows that telemedicine enhances continuity of care even during pandemics such as COVID-19.

The patients interviewed also applauded telemedicine for saving their time that would otherwise be wasted through traveling to healthcare facilities and waiting in long queues. This enhances the efficiency and convenience of healthcare facilities in responding to the patient's needs. One patient noted, "Consulting my doctor through a video call saves me the struggles of traveling to the healthcare facility, dealing with high traffic on the roads, and queuing for many hours, which would leave me completely exhausted by the end of the day. Telemedicine saves time and money and reduces the risks of infections, especially during flu season." [P.3] This feedback demonstrates how telemedicine not only provides logistical convenience but also enhances patient safety by minimizing exposure to crowded areas.

Some patients also regarded AI as a fundamental administrative tool that can be used to enhance administrative processes in the healthcare setting. The participants cited various healthcare administrative functions, such as workflow streaming, appointment scheduling, and prioritization of patients, which can be effectively handled by AI, ultimately enhancing the efficiency of services offered. One patient shared, "AI can help in the management of appointments and prioritization of cases. This enhances the efficiency of healthcare institutions, reducing delays in appointments and provision of services" [P.4] The sentiment expressed by this participant portrays AI as a crucial administrative tool that has the potential to streamline and automate key administrative functions in healthcare settings.

Overall, interviews conducted in this study portray AI and telemedicine as crucial tools for promoting flexible interactions between the healthcare provider and the patient; it is time-saving, cheaper, and convenient, especially for chronically ill patients whose immune systems have been greatly compromised, increasing the risks to contracting infections spread through human contact. It allows such patients to follow-up consultations at the convenience of their homes, as explained by one of the chronically ill patients interviewed in this study: "My ability to consult my doctor right from here has been a game changer for me, thanks to telemedicine. I rarely travel for consultations, which has greatly helped me remain more productive and efficiently plan my daily routines." [P.5] The above sentiments demonstrate the importance of telemedicine in provision of continuous, quality care, which leaves the patient satisfied.

**Impact on patient-doctor communication and relationships.** Participants, particularly patients, highlighted that integrating AI and telemedicine into healthcare has significantly enhanced the frequency and quality of communication between them and healthcare professionals. They argued that these technologies allow healthcare providers to customize communication to suit the patients, hence enhancing the personal connection between the patient and the doctor. Telemedicine's ability to personalize messages according to the patient's needs and preferences was considered one way of providing patient-centered care by the participants, which influenced their satisfaction levels with this technology. As expressed by one of the patients, "Engaging in a discussion with a physician who knows you at a personal level makes the patient feel good because it demonstrates that somebody knows you and cares about you and your condition and is determined to ensure you live a quality life. The personalization of communication between the patient and doctor creates a stronger bond, sense of satisfaction, trust, and confidence in the doctor." [P.3] The above insights highlight the importance of telemedicine in fostering meaningful and trustful relationships and communication, which enhances patient's experience and satisfaction with digital health technologies.

The participants equally acknowledged the importance of AI in improving patient-doctor communication experience. They argued that this technology enhances the access, use, analysis, and retrieval of patient records, leading to informed decisions and adequate preparation of healthcare providers for consultations. With adequate preparation and having all facts about the patients, including medical history, the doctors are able to clearly advise and communicate with patients, leading to better health outcomes. These findings were echoed by one of the patients, who remarked, "There is nothing as reassuring as knowing that your doctor knows your medical history; it is annoying to patients to repeatedly ask for details they shared a long time ago, some of which they have already forgotten and which are needed to make the correct clinical decisions. AI helps doctors to retrieve these records and even recommend medical interventions based on these records. This builds the patient's trust in the doctor and some level of satisfaction in the quality of care provided." [P.12]. Based on these sentiments, it is evidently clear that AI enhances doctor preparation and communication with the patient, consequently building trust and improving the patient's experience and satisfaction with the quality of care provided.

The patients acknowledged the significant role played by AI and telemedicine in monitoring the progress of patients and reaching out to them when there is a need. This monitoring ensures that the interventions are applied on time, preventing the health condition from deteriorating. One of the participants testified, "I receive regular updates about my health, including when to take or change the medication, what dietary and lifestyle choices I should embrace, all aiming to enhance my health outcomes. In case I notice peculiar symptoms or reactions to food, environment, or drugs, I inform my doctor, who calls me and gives appropriate recommendations. This digital health technology gives a sense of peace" [P.5] These proactive measures create a supportive health ecosystem that makes patients feel that their health needs are prioritized.

Telemedicine was frequently associated with convenience in communication, as patients valued the ability to consult healthcare providers without the need for physical visits. Therefore, the findings showed that telemedicine creates a strong bond between patients and doctors, providing a flexible communication and interaction platform that facilitates personalized, high-quality communication between patients and doctors, strengthening a sense of connection, trust, and satisfaction with the quality of care provided. In the words of one of the patients, "Telemedicine fosters a connection between patients and doctors, tranquility, and sense of worth" [P.4] Both AI and telemedicine complement each other, creating a supportive environment that is patient-centered, flexible, and customizable to patient's unique needs and preferences. The combination of the two digital health technologies creates a convenient, efficient, responsive, and accessible, patient-centered healthcare environment, which enhances positive health outcomes and a sense of satisfaction.

**Efficiency gains and ethical/privacy concerns.** Participants recognized that AI and telemedicine bring substantial efficiency gains in healthcare delivery by streamlining various processes, from booking appointments to managing follow-ups. In particular, they associated telemedicine with facilitating faster access to healthcare providers and saving time since there is no traveling and queuing for healthcare services. Chronically ill patients praised telemedicine for its convenience and for helping them avoid unnecessary visits to healthcare facilities, which not only consume their time and expenses but increase their exposure to infections. This response shows that telemedicine enhances efficiency by reducing the distance, time, and other logistical challenges associated with in-person visits.

The participants, especially patients, recognized the potential of AI in streamlining and enhancing the efficiency of administrative functions in healthcare settings. Some participants noted that the use of AI in scheduling and patient triage systems fastened the appointment

process in healthcare organizations, leading to more allocations of adequate time for consultations and ultimately improving the quality of patient-doctor interactions in clinical settings. In support of this, Participant P.12 asserted, "AI-scheduling is based on the type of health problem the patient is suffering from; hence, it allocates adequate time that reflects the health condition, thereby allowing adequate time for patient-doctor interaction during the consultation. This also eliminates paperwork and efficiency in queue management since patients can know at what time they will see the doctor." [P.12] These insights demonstrate that AI streamlines the administrative processes, creating adequate time for the provision of quality, personalized care to different patients.

Even though the participants appreciated AI and telemedicine for enhancing the efficiency and personalization of care provided, some raised various ethical and privacy concerns regarding these digital health technologies. Several participants expressed their discontentment with the handling and management of patient's sensitive health information, arguing that it is prone to data security breaches. For instance, Participant P.15 feared that "patients' personal data can be at risk of being accessed by unauthorized people, due to lack of strict measures and standards on how data managed via AI tools should be managed and stored." Such concerns demonstrate the need for implementing robust data security features to enhance patients' trust and confidence in electronic health records management systems.

Another group of patients raised some ethical concerns pertaining to the use of AI-based decision-making systems in clinical settings. They feared that AI-based decision-making systems lack human empathy, as witnessed in medical practitioners, which increases their likelihood of making wrong decisions that are inconsiderate of the unique patient's characteristics. In fact, one of the patients interviewed noted, "I don't like the idea of programmed machines making critical decisions about my health. My body is so complicated to be diagnosed or even treated by a machine that has been programmed by a person who does not know me or has interacted with me to understand my health and other unique characteristics. I want a human being with whom I can converse and get to know myself in person. Human touch is irreplaceable by AI." [P.3]. These sentiments emphasize the importance of human (patient-doctor) interactions even in technology-based clinical practice.

To sum it up, participants demonstrated that while adopting and using AI and telemedicine enhances the efficiency of healthcare services, its use in clinical settings also raises some data privacy and ethical concerns. These concerns underpin the importance of strengthening security protocols in digital health technologies to protect sensitive data from illegal access. The sentiments expressed by the participants demonstrated the need to balance efficiency and ethical considerations in order to enhance users' satisfaction and confidence in using AI and telemedicine in the healthcare sector. This aligns with P.9's concluding remarks, in which he asserted, "AI is beneficial and will be beneficial in the future of the healthcare sector. However, the full realization of its benefits necessitates the protection of patients' privacy and complementing it with human doctors to avoid the feeling of being treated by a programmed machine among many patients." These remarks demonstrate the need to make digital health technology a complement of medical experts rather than a replacement and address the ethical and privacy concerns associated with the adoption and use of digital health technologies such as AI and telemedicine.

## Discussion

### Summary of the findings

This research sought to understand the benefits, challenges, and enablers of successful adoption and utilization of AI and telemedicine in the UAE's healthcare setting. The findings

showed that the adoption and use of AI and telemedicine are associated with improved communication, accessibility, and convenience. Telemedicine has helped patients in remote areas access healthcare services in the comfort of their homes, reducing the need to travel, hence saving time and money, and avoiding logistical challenges associated with traveling to healthcare facilities, as well as protecting themselves from contracting infections associated with close interactions with other people. Healthcare professionals also associated AI with streamlining and improving administrative processes and helping healthcare professionals make data-informed decisions, which improve patient-doctor relationships and the efficiency of clinical care.

This study has also identified several obstacles to the adoption and effective use of AI and telemedicine in the UAE healthcare setting. The participants linked these technologies with risks of misinformation and misdiagnosis, serious safety and ethical concerns, financial and infrastructural limitations, internal resistance to adopt and use them, and skill gaps, which hindered the seamless incorporation of these technologies into clinical practices. The findings demonstrate that the full potential of AI and telemedicine can be realized by addressing these barriers to create a conducive, technology-friendly environment in the healthcare sector. The findings have, therefore, provided a critical understanding of the challenges and enablers of AI and telemedicine adoption and use in the healthcare sector and valuable insights on issues that need to be addressed to enhance their effectiveness in the healthcare setting.

## Comparison with existing literature

Most of this study's findings echo themes identified in digital health technologies scholarship conducted in other settings, which associate AI and telemedicine with the improvement of accessibility, reduction in healthcare costs, and improvement in convenience. For instance, available literature associates telemedicine with the enhancement of accessibility and efficiency, especially among chronically ill patients or people in remote locations, convenience, and reduction of traveling for in-person visits to healthcare facilities [4,5,11–13]. This benefit, which emerged strongly in our study, reflects global trends showing that patients appreciate the accessibility afforded by telemedicine, as seen in a literature review by Walley et al. [13] on patient satisfaction with telemedicine during COVID-19.

In terms of cost-effectiveness, our study echoes findings from previous research highlighting that digital health technologies can reduce both time and financial burdens for patients and healthcare providers. A study by Phuong et al. [5] showed that AI and telemedicine streamline processes such as patient scheduling and record-keeping, thereby decreasing administrative costs, a finding corroborated by participants in this study who noted similar advantages in the UAE context. However, our research adds a unique layer by revealing that in the UAE, these cost-saving benefits are especially valued due to the nation's high healthcare expenditure, which is driven by an increasing demand for advanced medical services. Therefore, this study supports global findings on cost savings while emphasizing specific economic motivators within the UAE.

Despite these parallels, our study presents some unique challenges and barriers that distinguish the UAE context from other regions. Unlike studies conducted in Western healthcare systems, where the integration of digital health technologies may face fewer structural limitations, participants in the UAE pointed to infrastructural shortcomings as a significant barrier to adoption. Undeniably, the UAE has made significant developments in its digital infrastructure; however, significant incompatibilities remain, especially in the integration of AI with healthcare providers' daily workflows. This varies greatly from European healthcare settings, where infrastructural integration of AI systems into the healthcare sector is a

non-issue [13]. Therefore, this study provides more contextual findings, showing that while UAE has achieved notable success in improving its digital infrastructure, its integration into the healthcare setting suffers from technical and logistical challenges, which hinder the effective widespread adoption and use of AI and telemedicine in the healthcare sector.

This study's findings on the resistance to change as a significant barrier to the adoption and use of digital health technologies such as AI and telemedicine align with the Technology Acceptance Model (TAM) proposed and tested in the existing literature [14–16]. Studies that have employed the TAM framework (which are mostly quantitative) have demonstrated a strong correlation between perceived usefulness and perceived ease of use and adoption of different technologies [14–16]. These findings are also echoed in the present study, which established that medical practitioners are hesitant to adopt and use AI and telemedicine because they fear that it might disrupt their routine workflows or they might lose control over clinical decisions. Similarly, a study by Tursunbayeva and Renkema's [17], which also investigated the adoption of AI in healthcare, supports these findings by positing that AI adoption in clinical settings is hindered by user's fear that it might affect their professional autonomy. While the present study echoes these findings, it demonstrates that in the UAE context, these fears are exacerbated by regional and cultural factors, hence demonstrating the need for a tailored approach to enhance and accelerate digital health technology acceptance in the healthcare sector.

Ethical and privacy concerns surrounding the adoption and use of digital health technology identified in this study resonate with the existing literature, especially in studies involving AI technology. For instance, Anyanwu et al. [18] and Khalid et al. [19] noted that their participants were also worried about data privacy in AI applications. This study, however, provides new insights, demonstrating that the absence of transparent data management frameworks for AI applications in the UAE healthcare setting exacerbates such security concerns. In support of these findings, a recent study by Khalid et al. [19] observed that the extent to which patients trust digital health technologies depends on the robustness and transparency of the privacy measures. While privacy concerns are a universal problem, especially in AI applications, this study suggests that it may be more pronounced in countries whose digital health policies are still developing, such as the UAE.

Lastly, the findings on patient-provider relationships show some contrasts with previous studies. Our study established that AI and telemedicine significantly influence the relationship between the healthcare provider and the patient. While this finding is supported by scholars such as Nathaniel and Therissa [20], who associated AI with strengthening the relationship between the healthcare provider and patient through streamlining data sharing and access to the patient's medical history, this study demonstrates a more complex interaction between digital health technology, patient, and healthcare provider. While some patients praised telemedicine for its convenience in enhancing their access to care in the comfort of their homes, others blamed it for lacking empathy and personal touch associated with physical, face-to-face doctor-patient interactions. The latter group of participants demonstrated the importance of empathy in determining the quality of care, health outcomes, and patient satisfaction. Such concerns may have been heightened by the unique UAE cultural context where close personal relationships are highly valued. These findings demonstrate that while AI and telemedicine enhance clinical efficiency, their implementation in the UAE context must consider the value of close personal relations in this society to be successful; it should strive to provide more personalized care and communication. The findings also suggest that the implementation of telemedicine and AI in the UAE should prioritize a hybrid approach, where digital convenience merges with close personal interaction with the patient to create personal connections and relationships, which are highly valued in this unique cultural context.

To sum it up, this study significantly contributes to the existing knowledge on the adoption of digital health technology by not only supporting the existing themes of technology adoption, such as ethical and privacy concerns, efficiency, and convenience, but also demonstrating how the unique UAE context heightens these themes. Our study shows that unique regional challenges, such as infrastructural challenges and cultural context, affect the effectiveness and efficiency of AI and telemedicine in the healthcare sector. This study shows that cultural resistance to new technologies, which is deeply entrenched in the country's traditional healthcare practices and the dynamic nature of the provider-patient relationship, significantly affects the acceptance, adoption, and use of telemedicine in the UAE, which varies from other contexts reported in the previous literature.

This study has demonstrated the importance of taking the contextual, cultural, and political factors into consideration in the implementation of digital health technologies, especially in a country like the UAE, where health practices are deeply intertwined with cultural values. This study provides a novel approach to technology adoption and implementation, demonstrating that the adoption and implementation framework must take a culturally sensitive approach to avoid hurdles that might limit the potential of such technologies. The findings have demonstrated that each region has unique factors, and hence, universal technology adoption models cannot be universally applied without a comprehensive understanding of the country's context. Therefore, this study demonstrates the need for future research to explore region-specific dynamics to develop a holistic framework for the adoption of digital health technologies such as AI and telemedicine to ensure that the healthcare sector benefits from these technologies.

## Strengths and limitations

Our study uniquely contributes to the global literature on the adoption and use of emerging digital health technologies such as AI and telemedicine by providing insights of the UAE healthcare practitioners and patients. A major strength of this study lies in its approach of presenting the perspective of both healthcare professionals and patients, which provides a comprehensive understanding of the implications of such technologies in clinical settings. The use of qualitative research design helped the study capture complex nuances that may have been overlooked by quantitative studies, which mostly test theories and models developed in other contexts. Therefore, using the qualitative methods to capture patients' and practitioners' insights has helped understand the complex and unique interactions between technology, patients, and practitioners in the UAE context, hence providing valuable insights on how specific regional factors such as healthcare infrastructure, political factors, and cultural attitudes might impact the adoption and use of digital health technologies like AI and telemedicine. Such insights not only contribute to the global discussion on technology adoption and acceptance but also helps local policymakers and stakeholders in the healthcare sector identify areas or concerns that they must address to accelerate the adoption and use of digital health technologies in clinical settings for better health outcomes.

Nonetheless, despite the above contributions, this study suffered some methodological limitations that may impact the generalizability of the findings. Being qualitative in nature, this study employed a sample of 15 interviewees, who, though they provide qualitative insights on the research problem, limit the extent to which this study's findings can be generalized to broader contexts. The generalization principle was also limited by the fact that the study was based on participants drawn from only two healthcare institutions in the UAE, implying that their views and experiences cannot be said to represent those of other practitioners and patients in other hospitals or regions in the country. The use of a purposive sampling strategy, where only individuals who had directly experienced and interacted with AI and telemedicine were interviewed,

also narrowed the range of perspectives as insights from those who have not interacted with these technologies were not consulted. This implies that the findings of this study cannot be applied to individuals who have not experienced or interacted with these digital health technologies.

## Conclusions

### Implications for practice and policy

This study's findings demonstrate that full realization of the potential of AI and telemedicine in the UAE healthcare sector necessitates adopting a framework that is considerate and sensitive to the country's unique infrastructural and cultural context. The study has also established that training and education are crucial factors to consider in the adoption and use of AI and telemedicine technologies as they boost the sense of self-efficacy, confidence, and competence, ultimately reducing internal resistance to change. This implies that healthcare institutions and the government have a role to play in ensuring that healthcare practitioners are equipped with the right technical skills and knowledge to effectively operate and integrate AI and telemedicine into clinical practice to enhance their efficiency.

The findings also underscore the importance of awareness initiatives in enhancing the acceptance and use of digital health technologies and minimizing resistance among patients and providers. Awareness initiatives help to eliminate misconceptions and inform users of the benefits of the technology, hence reducing opposition and accelerating the adoption of such technologies. This study also demonstrates the importance of developing standards, guidelines, and procedures for implementing AI and telemedicine in the UAE healthcare sector and addressing ethical and data security concerns raised by the participants. The findings, therefore, suggest the need for developing a national framework for digital health technology adoption that covers issues such as data management, patient management, training, and other aspects that enhance the adoption, acceptance, and use of technologies such as AI and telemedicine to improve the quality of care provided in the UAE healthcare institutions. The policies should also try to eliminate hurdles identified in this study through incentive programs and strategies.

### General recommendations for future research

Based on the abovementioned limitations, this study makes the following recommendations for future research. First, future research should recruit larger sample sizes from different hospitals in the country to capture a wide range of perspectives and enhance the generalizability of the findings to a broader context. Expanding the sample size and regional coverage enhances not only the diversity of perspectives but also helps to understand how cultural, political, and regional diversity influences the integration and effective use of AI and telemedicine.

Secondly, future studies are encouraged to consider clinical settings and practitioners who have not adopted or been exposed to AI and telemedicine. This inclusion would provide balanced insights into the factors influencing both acceptance and resistance to these technologies within the healthcare sector, capturing voices that may reflect hesitations, logistical challenges, or skepticism. Exploring these perspectives would contribute to fully understanding the barriers and concerns that delay digital health adoption. In addition, longitudinal studies are recommended to assess the sustained effects of AI and telemedicine on patient outcomes, healthcare provider satisfaction, and overall system efficiency. Such studies could reveal longer-term impacts and potentially unintended consequences, allowing policymakers and healthcare leaders to make well-informed, adaptive decisions about the future role of digital health technologies. Addressing these research gaps would create a stronger foundation for understanding how AI and telemedicine can best be integrated into healthcare settings across diverse contexts.

## Overall conclusion

This study provides valuable insights into the contextual factors influencing the adoption and effective use of digital health technologies within the UAE healthcare system, specifically AI and telemedicine. This research captures the perspectives of healthcare professionals and patients, thereby shedding light on the benefits, challenges, and facilitators of these technologies. This study's findings demonstrate that the adoption and use of AI and telemedicine enhance clinical efficiency, accessibility, and convenience, while on the other hand, it is associated with some serious safety and privacy concerns, fears of misinformation and misdiagnosis, and faced by limited infrastructural capabilities. The study has established that infrastructural gaps and cultural and political factors can be a significant hindrance to the adoption of AI and telemedicine, underscoring the importance of adopting a culturally sensitive approach to the integration of digital health technologies in the UAE. The findings of this study not only echo the global literature on digital health technology adoption but also identify unique aspects of the UAE healthcare sector that need a context-specific approach to realize the full potential of these technologies.

Therefore, this study significantly contributes to nursing scholarship and practice as it demonstrates the challenges and facilitators of integrating digital health technology in clinical practice. The study provides guidance to policymakers and healthcare practitioners on how to streamline and optimize the integration of AI and telemedicine into the healthcare sector. The study also demonstrates the importance of training and educational programs, public awareness initiatives, and consideration of the cultural and political environment in the development and adoption of digital health technologies in different healthcare settings. The study therefore highlights the importance of employing a balanced approach in the integration of AI and telemedicine in the healthcare sector to ensure it provides patient-centered care that aligns with the prevailing social, cultural, and political context.

**Specific policy recommendations.** Seamless incorporation of AI and telemedicine into the UAE healthcare sector necessitates consideration of the following issues by healthcare policymakers and stakeholders: (1) investment in flexible and comprehensive training and educational programs to equip healthcare professionals with relevant technical skills to enable them to work with these technologies; (2) creating public awareness on the benefits and limitations of the AI and telemedicine to accelerate acceptance and adoption and reduce cultural resistance; (3) incentivizing the adoption and use of AI and telemedicine by particularly focusing on major hindrances to the adoption to enhance access and equity in healthcare, and; (4) establishment of clear legal frameworks and policies addressing ethical and privacy issues associated with management and handling patients' data digitally to create trust, confidence, and enhance satisfaction among patients.

## Supporting information

**S1 File. Interview transcripts.**
(DOCX)

## Acknowledgments

The author acknowledges and appreciates all participants (patients and healthcare professionals) who voluntarily participated in this study. Without your invaluable contributions and time during the data collection and validation process, this study would not have been successful. Thank you so much.

## Author contributions

**Conceptualization:** Azza Alkaabi.

**Data curation:** Azza Alkaabi.

**Investigation:** Azza Alkaabi.

**Methodology:** Azza Alkaabi.

**Writing – original draft:** Azza Alkaabi.

**Writing – review & editing:** Deena Elsori.

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
