## [Decision Letter · Decision Letter 0]

11 Oct 2024

PDIG-D-24-00287

Navigating Digital Frontiers in Healthcare: A Qualitative Exploration of Healthcare Professionals' and Patients' Experiences with AI and Telemedicine

PLOS Digital Health

Dear Dr. Alkaabi,

Thank you for submitting your manuscript to PLOS Digital Health. After careful consideration, we feel that it has merit but does not fully meet PLOS Digital Health's publication criteria as it currently stands. Therefore, we invite you to submit a revised version of the manuscript that addresses the points raised during the review process.

Please submit your revised manuscript within 60 days Dec 10 2024 11:59PM. If you will need more time than this to complete your revisions, please reply to this message or contact the journal office at digitalhealth@plos.org. Please include the following items when submitting your revised manuscript:

We look forward to receiving your revised manuscript.

Kind regards,

Shannon Freeman, PhD

Academic Editor

PLOS Digital Health

Shannon Freeman

Academic Editor

PLOS Digital Health

Journal Requirements:

1. We ask that a manuscript source file is provided at Revision. Please upload your manuscript file as a .doc, .docx, .rtf or .tex.

Additional Editor Comments (if provided):

Thank you for submitting your paper "Navigating Digital Frontiers in Healthcare: A Qualitative Exploration of Healthcare Professionals' and Patients' Experiences with AI and Telemedicine" for consideration in PLoS Digital Health. The reviewers have provided comprehensive and detailed feedback for the authors. The authors are encouraged to carefully consider the suggestions, especially those from reviewer 2. There are concerns about the methodology and whether saturation has been met given the limited number of participants and the breadth of topic addressed. Further attention to these two areas specifically will enhance the quality of the paper.

Reviewers' comments:

Reviewer's Responses to Questions

**Comments to the Author**

1. Does this manuscript meet PLOS Digital Health’s publication criteria ? Is the manuscript technically sound, and do the data support the conclusions? The manuscript must describe methodologically and ethically rigorous research with conclusions that are appropriately drawn based on the data presented.

Reviewer #1: Yes

Reviewer #2: Partly

Reviewer #3: Yes

2. Has the statistical analysis been performed appropriately and rigorously?

Reviewer #1: N/A

Reviewer #2: N/A

Reviewer #3: Yes

3. Have the authors made all data underlying the findings in their manuscript fully available (please refer to the Data Availability Statement at the start of the manuscript PDF file)?

Reviewer #1: Yes

Reviewer #2: No

Reviewer #3: Yes

4. Is the manuscript presented in an intelligible fashion and written in standard English?

PLOS Digital Health does not copyedit accepted manuscripts, so the language in submitted articles must be clear, correct, and unambiguous. Any typographical or grammatical errors should be corrected at revision, so please note any specific errors here.

Reviewer #1: Yes

Reviewer #2: Yes

Reviewer #3: Yes

5. Review Comments to the Author

Please use the space provided to explain your answers to the questions above. You may also include additional comments for the author, including concerns about dual publication, research ethics, or publication ethics. (Please upload your review as an attachment if it exceeds 20,000 characters)

Reviewer #1: The manuscript is incredibly interesting and presents novel and useful data to the field of digital health. 

The authors have done a wonderful job in displaying their comprehensive research to the reader. Coming from the qualitative research field myself, I have quite enjoyed reviewing this paper. I have added my suggestions/ changes in the manuscript. Along with that, I would like to point out that the manuscript needs to be adapted to this journal's standards, e.g. formatting, adding line numbers, front page, font sizes and the overall structure etc. (an example can be found on the website under submission requirements).

Reviewer #2: Many thanks for allowing me to review this article. I have a few suggestions that authors may want to consider in order to improve this report: 

1. Please insert subheadings into the abstract.

2. Because the study was conducted in UAE, this should be reflected in the title and abstract.

3. Because this was qualitative research, the authors should clearly state that they explored ‘perceptions of’ or ‘attitudes towards’ technologies rather than saying they examined the effectiveness. It is challenging to assess effectiveness through qualitative research. 

4. I am not persuaded that combining AI and telemedicine together in 15-interview research with two different groups is a robust enough methodology. AI and telemedicine are two distinct and very broad fields, and each deserves their own investigation. I am concerned to what extent these two concepts were used interchangeably throughout these interviews.

5. Looking at the overall type and the number of publications, I am not convinced that the authors provide a comprehensive overview of the study rationale and synthesis. There are already multiple reviews of literature looking at AI and telemedicine acceptability that are not mentioned in this study. 

6. I am not persuaded that the study reached saturation with 8 HCPs and 7 patients due to the range of topics discussed on the subjects. The authors make no mention of any data saturation; they only mention that 15 out of 30 participants agreed to take part in the study. As such, I am concerned that the study could be incomplete and the analysis rather superficial without generating any new knowledge. 

7. Also, the cohort was above the age of 34 (mostly in their 40s), which could skew the analysis through the lack of younger participants with more positive views on telemedicine.

8. Although the researchers claim they didn’t use any specific theory for their analysis, the theme of barriers and facilitators is usual in implementation sciences. Thus, to what extent the research was designed to identify barriers and facilitators for AI/telemedicine? 

9. How did the researchers aim to reduce their subjectivity in data analysis? Was the analysis conducted by one person? What was their level of expertise in conducting qualitative data analysis?

10. Was the study approved by any research ethics committee? I cannot see any reference number to the ethics clearance. 

11. The results section has a description of questions asked to participants, which will be better suited in the methods section than in the results.

12. I am concerned that authors may overinterpret some of the data analysis because they were making a claim that might not be fully grounded in the qualitative data. For example, the results section should always refer to participants’ perceptions “Participants thought/saw/believed/viewed/suggested/agreed/felt etc” rather than making a substantial claim about AI (e.g. “ It [telemedicine] also allows them to give more intensive care to the patients who require such care because their needs can be assessed more comprehensively.”)

13. The presentation of the quotes is also without the authors’ interpretation. It is almost as if the quotes are inserted for the readers’ interpretation. Instead, a careful description of the data analysis outcomes should be outlined with quotes to demonstrate the interpretations. Some quotes even formulate a stand-alone paragraph, which is unacceptable in qualitative research write-ups. 

14. Reading through the analysis, it also does not separate AI from telemedicine, and while these two technologies are very different in terms of their applications, it isn't easy to understand what participants were referring to. The analysis is lacking sufficient depth in that respect. 

15. There is an unnecessary level of repetitiveness throughout the manuscript. For example, there is no need to repeat the study aims at the beginning of the discussion. Instead, the authors should reiterate the main findings and explain them. 

16. As many of the themes reflect ideas previously identified in other studies, such as cost or convenience, the discussion section would benefit from the outline of the new or novel knowledge captured in this study that had not been previously uncovered. I think that would make this study unique and citable. Thus, more attention should be dedicated to the unique and odd findings in the study rather than repeating what was already reported in other qualitative studies and systematic reviews. 

17. In the discussion – strengths and weaknesses the authors claim “This study’s primary strength is being the first to comprehensively investigate the contextual factors influencing the adoption and effective use of digital health technologies like AI and telemedicine in clinical settings.” However, this is not the first qualitative study to investigate this. [Young, A. T., Amara, D., Bhattacharya, A., & Wei, M. L. (2021). Patient and general public attitudes towards clinical artificial intelligence: a mixed methods systematic review. The lancet digital health, 3(9), e599-e611.] The authors may want to review the existing literature on the subject to reassess their novelty. 

18. I agree with the authors about the study limitations and the narrow focus on the UEA. I would suggest rewriting the report to focus on sufficiently covering the UEA context in the introduction, title, and abstract so that researchers can quickly identify its scope. I would also recommend adjusting the discussion section to fit it within the UEA context so that others can understand how the geo-political issues influence those perceptions. I think a major rewrite is therefore needed so that this article is carefully placed within its healthcare context. 

19. The is also a need for a stronger ‘implications’ subsection from this study outlining how this knowledge can be applied in clinical settings.

Reviewer #3: A. SUMMARY OF THE RESEARCH FINDINGS AND OVERALL IMPRESSION

i. Summary of the Research findings

Abstract section

The abstract effectively summarizes the study, providing a clear outline of the research objectives, methodology, and key findings. It highlights the benefits of AI and telemedicine, as well as the barriers and facilitators affecting their adoption. However, the abstract could be improved by including specific details about the study's location and the unique contribution of the research to existing literature, which would provide readers with a clearer context. Future work and the policy implications from the findings are also missing and need to be included

Other Sections

The introduction section has provided a comprehensive background on digital health technologies and their impact on healthcare. It sets the stage well by discussing AI and telemedicine's potential and the paradigm shift they represent in healthcare delivery. However, the section could be improved by specifying the research gap more explicitly and clearly stating how this study addresses it. 

Looking at the methods section, it is well-detailed, describing the qualitative descriptive research design, sampling procedure, and data collection methods. The use of semi-structured interviews and thematic analysis is appropriate for the study's objectives. However, the study's limitations in sampling, such as the small sample size and focus on only two Emirates in the UAE, could be highlighted earlier in this section. Including a justification for choosing a qualitative approach over a mixed-methods approach could also strengthen the methodological rigor.

The results section has provided a thorough presentation of the findings, organized around the study's key objectives. Themes are well-defined, and quotes from participants effectively illustrate the insights gathered. The section successfully captures the complexities of healthcare professionals' and patients' experiences with AI and telemedicine. However, some of the findings, particularly those related to specific barriers, could be more critically examined to explore deeper implications.

The discussion section interprets the results well, linking them to existing literature and highlighting the practical implications of the findings. It effectively addresses the benefits and drawbacks of AI and telemedicine and provides insights into the barriers and facilitators of their adoption. However, the discussion could be enriched by exploring the findings' theoretical implications more deeply and comparing the study results with those from different contexts outside the UAE.

The conclusion concisely summarizes the main findings and offers practical recommendations to improve the adoption and effectiveness of AI and telemedicine. However, it would be beneficial to suggest specific actions or policy changes needed to address the identified barriers, such as targeted training programs or infrastructural investments.

Overall Impression

The manuscript provides valuable insights into the contextual factors influencing the adoption of AI and telemedicine in clinical settings. It makes a significant contribution by exploring the perspectives of both healthcare professionals and patients. The thematic approach used in the analysis is appropriate and allows for a nuanced understanding of the challenges and opportunities associated with digital health technologies. 

B. SPECIFIC AREAS OF IMPROVEMENT

Add missing information in the abstract

There is need for more clarity in stating the research gap in the introduction section. The authors should more explicitly identify the research gap and clearly articulate how this study fills that gap compared to existing studies.

Readers would like to see some theoretical implications of the findings, particularly in relation to existing models of technology adoption in healthcare.

The conclusion should provide more specific policy recommendations based on the findings, such as initiatives to improve training, address infrastructural deficits, or develop legal frameworks to protect data privacy.

6. PLOS authors have the option to publish the peer review history of their article (what does this mean? ). If published, this will include your full peer review and any attached files.

**Do you want your identity to be public for this peer review?** For information about this choice, including consent withdrawal, please see our Privacy Policy .

Reviewer #1: No

Reviewer #2: No

Reviewer #3: No

---

## [Decision Letter · Decision Letter 1]

20 Feb 2025

Navigating Digital Frontiers in UAE Healthcare: A Qualitative Exploration of Healthcare Professionals' and Patients' Experiences with AI and Telemedicine

PDIG-D-24-00287R1

Dear Dr. Alkaabi,

We are pleased to inform you that your manuscript 'Navigating Digital Frontiers in UAE Healthcare: A Qualitative Exploration of Healthcare Professionals' and Patients' Experiences with AI and Telemedicine' has been provisionally accepted for publication in PLOS Digital Health.

Best regards,

Shannon Freeman, PhD

Academic Editor

PLOS Digital Health

**Additional Editor Comments (if provided):**

**Reviewer Comments (if any, and for reference):**

Reviewer's Responses to Questions

**Comments to the Author**

1. If the authors have adequately addressed your comments raised in a previous round of review and you feel that this manuscript is now acceptable for publication, you may indicate that here to bypass the “Comments to the Author” section, enter your conflict of interest statement in the “Confidential to Editor” section, and submit your "Accept" recommendation.

Reviewer #1: All comments have been addressed

Reviewer #3: All comments have been addressed

2. Does this manuscript meet PLOS Digital Health’s publication criteria ? Is the manuscript technically sound, and do the data support the conclusions? The manuscript must describe methodologically and ethically rigorous research with conclusions that are appropriately drawn based on the data presented.

Reviewer #1: Yes

Reviewer #3: Yes

3. Has the statistical analysis been performed appropriately and rigorously?

Reviewer #1: N/A

Reviewer #3: Yes

4. Have the authors made all data underlying the findings in their manuscript fully available (please refer to the Data Availability Statement at the start of the manuscript PDF file)?

Reviewer #1: Yes

Reviewer #3: Yes

5. Is the manuscript presented in an intelligible fashion and written in standard English?

Reviewer #1: Yes

Reviewer #3: Yes

6. Review Comments to the Author

Reviewer #1: Thank you for addressing my comments accordingly.

- I would like to raise the question, whether it is necessarily needed to include the line of questioning under the method section to such extent, from my point of view it would be enough to include it in the appendix/SI.

- Secondly, setting quotes apart from the written text by using a cursive font f.e. may improve readability.

Reviewer #3: All the comments (those i gave and from other reviewers) have been satisfactorily responded to.

I am confident that now this article can move to another level

7. PLOS authors have the option to publish the peer review history of their article (what does this mean? ). If published, this will include your full peer review and any attached files.

**Do you want your identity to be public for this peer review?** For information about this choice, including consent withdrawal, please see our Privacy Policy .

Reviewer #1: No

Reviewer #3: No
